

# Discrimination of foreign bodies in quinoa (*Chenopodium quinoa* Willd.) grains using convolutional neural networks with a transfer learning approach

Himer Avila-George[1], Miguel De-la-Torre[1], Jorge Sánchez-Garcés[2], Joel Jerson Coaquira Quispe[2], Jose Manuel Prieto[2,3] and Wilson Castro[4]

[1] Departamento de Ciencias Computacionales e Ingenierías, Universidad de Guadalajara, Ameca, Jalisco, México
[2] Facultad de Ingeniería y Arquitectura, Universidad Peruana Unión, Juliaca, Puno, Perú
[3] Escuela de Doctorado en Ciencia, Ingeniería y Gestión Alimentaria, Universidad Politécnica de Valencia, Valencia, Valencia, Spain
[4] Facultad de Ingeniería en Industrias Alimentarias, Universidad Nacional de Frontera, Sullana, Piura, Perú

## ABSTRACT

The rising interest in quinoa (*Chenopodium quinoa* Willd.) is due to its high protein content and gluten-free condition; nonetheless, the presence of foreign bodies in quinoa processing facilities is an issue that must be addressed. As a result, convolutional neural networks have been adopted, mostly because of their data extraction capabilities, which had not been utilized before for this purpose. Consequently, the main objective of this work is to evaluate convolutional neural networks with a learning transfer for foreign bodies identification in quinoa samples. For experimentation, quinoa samples were collected and manually split into 17 classes: quinoa grains and 16 foreign bodies. Then, one thousand images were obtained from each class in RGB space and transformed into four different color spaces (L\*a\*b\*, HSV, YCbCr, and Gray). Three convolutional neural networks (AlexNet, MobileNetv2, and DenseNet-201) were trained using the five color spaces, and the evaluation results were expressed in terms of accuracy and F-score. All the CNN approaches compared showed an F-score ranging from 98% to 99%; both color space and CNN structure were found to have significant effects on the F-score. Also, DenseNet-201 was the most robust architecture and, at the same time, the most time-consuming. These results evidence the capacity of CNN architectures to be used for the discrimination of foreign bodies in quinoa processing facilities.

## INTRODUCTION

Quinoa (*Chenopodium quinoa* Willd.) is a pseudocereal seed crop traditionally grown in the Andean region (*Shi et al., 2020*; *Nascimento et al., 2014*). According to the Food and Agriculture Organization of the United Nations (FAO), this grain is one of the superfoods

Corresponding author
Wilson Castro, wcastro@unf.edu.pe

that could provide food security in the 21st century (*Nowak, Du & Charrondière, 2016*). However, it is necessary to ensure adequate quality levels to enter new markets. In this respect, one of the most important criteria for commercialization, which establishes the national and international prices, is the percentage of foreign bodies (FB) (*Lecca-Pino et al., 2021*). FB refers to pits and pit fragments, unwanted seeds, bones, bone fragments, sticks, rocks, metals, plastics, and all unwanted objects in the food (*Toyofuku & Haff, 2012*); these elements can be a food safety hazard and affect a company's brand reputation (*Demaurex & Sall, 2014*).

Because of the variable nature of FB, a wide range of solutions have been devised to detect and remove them (*Toyofuku & Haff, 2012*); these solutions use the differences between *FB* and product, *e.g.*, (a) size or weight, (b) shape or color analysis, and (c) interaction with some part of the electromagnetic spectrum (*Graves, Smith & Batchelor, 1998*). According to *Edwards (2014)*, the principal techniques are those based on optical inspection, which uses forms, color, and shapes for discrimination. However, as explained by *Wu et al. (2019)* and *Moses et al. (2022)*, using a human expert for visual analysis in discrimination analysis is highly time-consuming, error-prone, and difficult to automate (due to that features require manual extraction).

In this context, new image analysis methods based on deep learning techniques, particularly convolutional neural networks (CNNs), are quickly becoming the preferred methods to address application-specific challenges such as image processing or classification tasks (*Too et al., 2019*; *Furtado, Caldeira & Martins, 2020*). Unlike traditional patterns recognition techniques, CNNs extract higher-level features progressively from the input (image, video, text, or sound) and incorporate feature selection into the learning process by updating their parameters and connections as a function of the error on a set of training data (*Han & Kuo, 2018*; *Wu et al., 2019*).

In the agricultural field, CNNs have been successfully applied in crop variety identification (*Too et al., 2019*), haploid and diploid seeds (*Altuntaş, Cömert & Kocamaz, 2019*), nematodes (*Abade et al., 2022*), plant disease recognition (*Too et al., 2019*), damage in milled rice grains (*Moses et al., 2022*), broccoli head quality discrimination (*Blok et al., 2022*), crop pest (*Ayan, Erbay & Varçın, 2020*), microstructural elements discrimination (*Castro et al., 2019b*), characterization of emulsions (*Lu et al., 2021*), among others. Using different common CNNs architectures (AlexNet, resNet, MobileNet, Inception, VGG16, DenseNet, among others), new architectures, and-or training approaches.

However, CNNs have not been widely used for foreign bodies' discrimination in food materials, being the work presented by *Rong, Xie & Ying (2019)*, the newest one focused on peanut crops. Consequently, in this article, it was evaluated three common CNNs under the learning transfer approach for foreign body discrimination in quinoa grains; then, we can summarize the contribution of this article as follows: (1) A primary images dataset of quinoa grains and different classes of foreign bodies was built, developing else a chart for visual discrimination of the main elements in quinoa grains; (2) augmented images dataset of quinoa grain and their foreign bodies were generated from the original dataset, and

different color spaces were obtained; and (3) three frequently used CNN were trained using training transfer methodology, and their statistical metrics were calculated and compared.

## METHODOLOGY

Figure 1 summarizes the methodology followed in the present work, which consists of the following steps: (a) manual selection of quinoa grains and foreign bodies, (b) images acquisition, (c) preprocessing, (d) CNNs training, and (e) models comparison.

### Quinoa sample

The quinoa grains were provided for the *Quinuanchis* cooperatives from the Caminaca district, Puno, Peru, corresponding to the 2018-2019 agricultural season. The grains covered the commercial white medium quinoa (1.5 to 2 mm). A sample of five kilograms was picked up and stored in polypropylene bags of capacity and transported to Universidad Peruana Unión facilities, where the selection of impurities and image acquisition was made.

### Manual selection

Different elements were manually separated and classified from the sample according to the Peruvian Technical Norm 205.062-2014 and the Codex Alimentarius Standard 333-2019. To make the classification easy, a chart based on the previously mentioned standards was developed to make this process (see Fig. 2).

This chart divides the possible elements into five main groups:

1. Good grains. According to the norms, those grains are complete and represent the quinoa variety's representative color.
2. Contrast grains. In this case, the color does not correspond to the variety.
3. Faulty grains. These grains correspond to the variety that presents some anomalies as broken, covered, and others.
4. Organic material. There are materials generated in the grain harvest and those due to biological agents (insect, rodent excrement, among others).
5. Inorganic materials. stones and clods picking join to the grain during the harvest process are present in this class.

Then, one thousand elements covering each class were selected from the sample and stored for the next steps.

### Images acquisition

An image of the thousand elements of the previously selected class was acquired using a digital microscope model *PCE-MM 800* from PCE America, Inc., Jupiter, FL, USA. It was connected to a laptop with a processor Intel I5, 128 MB RAM, and an Interface Card: USB 2.0 controlled by the *AMCap* software, version 9.016 from Genesys Logic, Inc. (San Jose, CA, USA). The lens was placed at 15 mm from the sample, acquiring one element per time and saving it in JPG format. The dataset is available at Zenodo, DOI: 10.5281/zenodo.7384664.
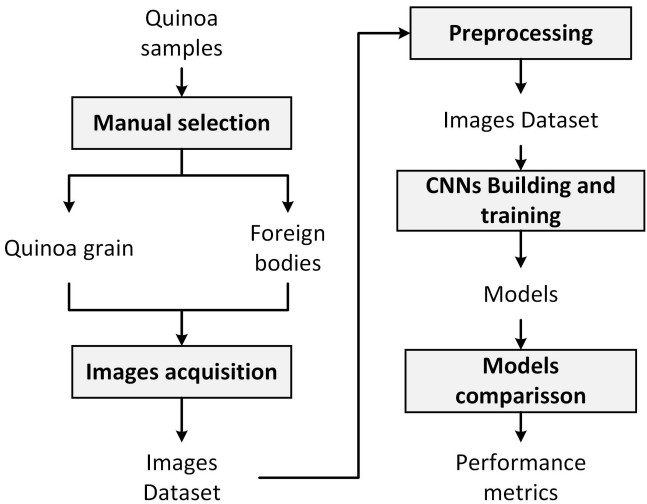

**Figure 1  The proposed methodology.**

## Preprocessing

The images in the dataset were pretreated, according to the proposals of *Altuntaş, Cömert & Kocamaz (2019)* and *Castro et al. (2019a)*, in order to be used as input in the CNN training; the steps are detailed below:

(a)  Image resizing. Images in the dataset were resized to [224 × 224] and [227 × 227] pixels per channel using the bi-cubic interpolation method.

(b)  Color space conversion. The original dataset, whose format was RGB of eight bits, was converted into four color spaces, details in Table 1. For color space conversion, the functions *rgb2lab*, *rgb2hsv*, *rgb2ycbcr*, and *rgb2gray* of the Image Processing Toolbox of MATLAB 2019a were used.

## CNNs building and training
### General structure of CNNs

CNNs are defined by *Abade et al. (2022)* as a class of machine learning models that are currently state-of-the-art in many computer vision tasks, applied to different kinds of images. To manage its applications, CNNs consist of a set of layers known as convolution, pooling, dropout, and fully connected layers and, in some cases *softmax* layers, see Fig. 3.

About basis structure of CNN, *Chen et al. (2020)* explains that the convolution layer is the basic brick, which allows discrimination among classes by extracting a small subset of spatially connected pixels in the input image channels by mean of convolution kernels, see Eq. (1).

$$H_i = \varphi(H_{i-1}W_i + b_i), \tag{1}$$

where $H_i$ is the feature map of the $i$th layer, $H_{i-1}$ is the convolution feature of the previous layer ($H_0$ is the original image), $W_i$ is the weight of the $i$th layer, $b_i$ is the displacement vector of the $i$th layer, and $\varphi(\cdot)$ represents the rectified linear unit (ReLU) function.

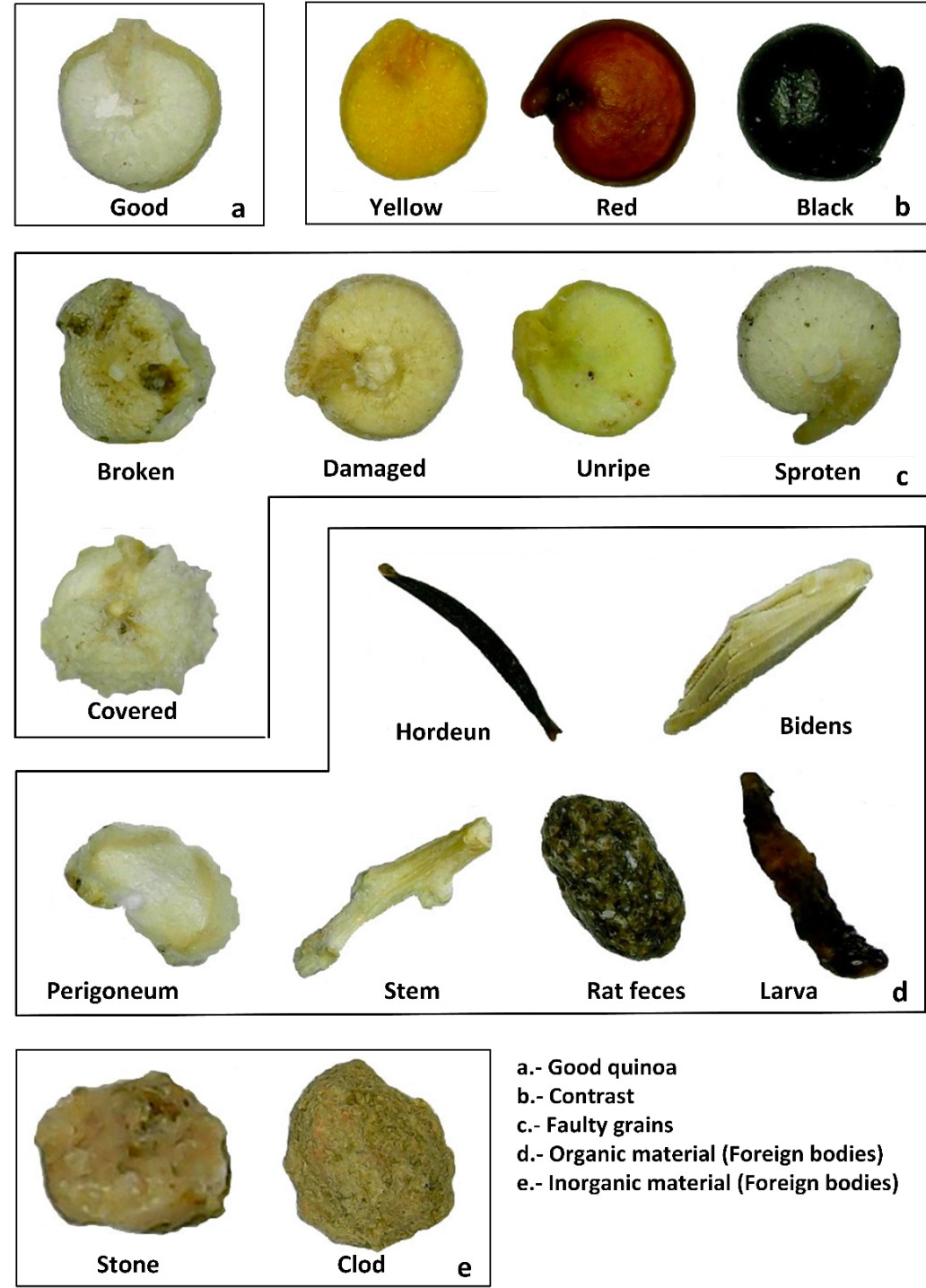

**Figure 2** **Possible material classes present in quinoa sample according to the normative.** (A) Good quinoa grain, (B) image contrasts, (C) faulty grains, (D) foreign bodies (organic material), and (E) foreign bodies (inorganic material).

**Table 1  Color spaces used in the study.**

| Space | Parameter | Range |
|---|---|---|
| | R | Red in the digital image [0, 255] |
| RGB | G | Green in the digital image [0, 255] |
| | B | Blue in the digital image [0, 255] |
| | L$^*$ | Luminosity derived from RGB [0, 100] |
| L$^*$a$^*$b$^*$ | a$^*$ | Red/green opposing colors [−128, 128] |
| | b$^*$ | Yellow/blue opposing colors [−128, 127] |
| | H | Hue derived from RGB [0, 360] |
| HSV | S | Saturation derived from RGB [0, 100] |
| | V | Value derived from RGB [0, 100] |
| | Y | Luminance derived from RGB [16, 235] |
| YCbCr | Cb | Blue minus luma derived from RGB [16, 240] |
| | Cr | Red minus luma derived from RGB [16, 240] |
| | I | Intensity derived from RGB [0, 255] |
| GGG$^*$ | I | Intensity derived from RGB [0, 255] |
| | I | Intensity derived from RGB [0, 255] |

**Notes.**
*GGG space is a pseudo color space in which each channel contains intensity values.

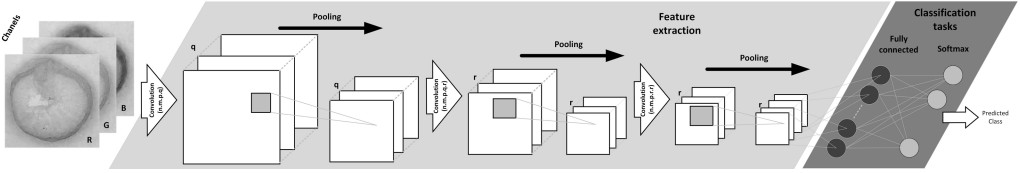

**Figure 3  Typical layers in a CNN.**

Due to that, convolution tasks increase the computational needs pool, and layers simplify the spatial dimensions of the information, averaging, normalizing, or extracting maximum values, see Eq. (2) (*Chen et al., 2020*).

$$x_j^l = down(x_j^{l-1}, s),  \qquad (2)$$

where $down(\cdot)$ is the down-sampling function, $x_j^{l-1}$ represents the feature vector in the previous layer, and $s$ is the pooling size. Another common layer is the fully connected (FC), which uses the extracted features for image classification. This task is usually performed by means of the *softmax* function; see Eq. (3).

$$softmax(z)_j = \frac{e^{z_j}}{\sum_{k=1}^{K}} e^{z_k} (for \, j = 1, \dots, k),  \qquad (3)$$

where $K$ = dimension of the $z$ vector.

In our study, three commonly used CNNs were selected, which ones are commented on below:

- AlexNet. Maybe the most known CNN, it has five convolution layers and three fully connected layers and introduced the use of *ReLu* function as an improvement to common

*sigmoid* or *tanh* functions, and 62.3 Millions of parameters (*Krizhevsky, Sutskever & Hinton, 2012*; *Altuntaş, Cömert & Kocamaz, 2019*; *Furtado, Caldeira & Martins, 2020*; *Lu et al., 2021*).

- MobileNet-V2. It is one of the most optimal networks with a small volume and few parameters (4.2 million parameters); this improves the accuracy of learning of spatial point inter-channel relationship (*Sandler et al., 2018*; *Chen et al., 2021b*; *Chen et al., 2021a*).

- DenseNet-201. Its architecture is composed of three types of blocks; a convolution block, a dense block, and a transition block, which connects two contiguous dense blocks; this produces a structure with 40 million of parameters (*Huang et al., 2017*; *Abade et al., 2022*; *Too et al., 2019*).

### *Transfer learning*

One of the advantages of CNNs is their ease of retraining to perform new specific tasks; the most frequent forms of CNN training are the following (a) Training from scratch and (b) Transfer learning. Training from scratch requires a considerable number of samples (images in our case) and a lot of time to perform the process (*Chen et al., 2021a*). On the other hand, transfer learning is a family of methods commonly referred to as fine-tuning; one of the most popular transfer learning approaches is to fine-tune all weights of a pre-trained model, with the last fully connected layer being replaced and randomly initialized in a new classification task. In fact, it is possible to fine-tune only a few layers, which are usually the last layers corresponding to a higher abstraction level, which turns out to be less costly in time and number of samples to use (*Ayan, Erbay & Varçın, 2020*; *Koklu, Cinar & Taspinar, 2021*; *Lu et al., 2021*; *Duong et al., 2020*), see Fig. 4.

According to *Chen et al. (2020)*, *Chen et al. (2021b)* and *Altuntaş, Cömert & Kocamaz (2019)* transfer learning method could be divided into the following steps:

1. Transfer the weight $(W_1, \ldots, W_n)$ and parameters of a trained CNN; in our case, the Deep Network Designer toolbox provides the well-trained CNNs for the next steps.
2. New CNN structure establishing, modifying (replacing or inserting) the last layers where classification tasks are done, according to the specific task we want.
3. Fine-tuning with the images dataset correctly labeled, the new CNN is trained to minimize the loss function $\zeta$ mentioned in (4).

$$\zeta = \frac{1}{|X|} \sum_i^{|X|} ln(P(y^i|X^i)), \qquad (4)$$

where $|X|$ is the number of training images, $X^i$ is the $i$th training image in the class $y^i$, $P(y^i|X^i)$ is the probability to be accurately labeled.

## Classifiers comparison

According to *Abade et al. (2022)*, the performance of classifiers was evaluated based on a $[17 \times 17]$ confusion matrix (CM) of real and predicted classes.

In a first step, in concordance with *Castro et al. (2019a)*, where determined the number of elements correctly and incorrectly classified; then, the following four measures were determined: ($TP_i$) true positives, ($TN_i$) true negatives, ($FP_i$) false positives, and ($FN_i$) false
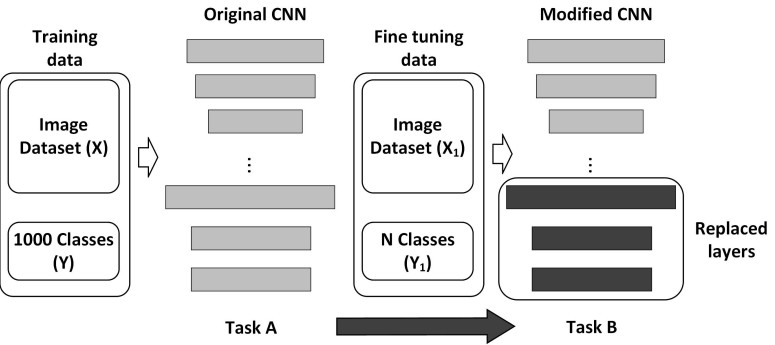

**Figure 4** General scheme of CNN transfer learning.

negatives. Next, performance metrics were calculated, where *n* represents the dataset size; see Eqs. (5) to (9). Then, the model's performance was summarised in *F-Score* value, being used to compare models.

$$Accuracy\,(ACC) = \sum_{i=1}^{n} \frac{TP_i + TN_i}{TP_i + TN_i + FP_i + TN_i}, \tag{5}$$

$$Precision\,(PRE) = \frac{\sum_{i=1}^{n} TP_i}{\sum_{i=1}^{n} TP_i + FP_i}, \tag{6}$$

$$Recall\,(REC) = \frac{\sum_{i=1}^{n} TP_i}{\sum_{i=1}^{n} TP_i + FN_i}, \tag{7}$$

$$Especificity\,(ESP) = \frac{\sum_{i=1}^{n} TN_i}{\sum_{i=1}^{n} TN_i + FN_i}, \tag{8}$$

$$F - Score\,(FSC) = 2 \times \left( \frac{PRE \times REC}{PRE + REC} \right). \tag{9}$$

Finally, each combination CNN—color space was trained thirty times, in a *hold-out* cross-validation strategy, and *F-score* saved in concordance with the performed by *Chen et al. (2020)* dividing 70% for training and 30% for validation. According to *Miraei Ashtiani et al. (2021)*, the reason for splitting the dataset into two subsets is that in small datasets, the additional split might lead to a smaller training set which may be exposed to over-fitting. The effect of variables in *F-score* was considered when $P_{value} < 0.05$, comparing it through *Tukey*-test.

## Software and hardware system description

The experiments were implemented in a workstation with Windows 11 Pro 64 bits, and Intel®Xeon®Gold 5218 processor, and an NVIDIA®Quatro RTX 6000 and 128 GB of

**Table 2   Parameters for transfer learning training in the study.**

| Parameters | Values |
| --- | --- |
| Back-propagation algorithm | SGDM |
| Max epochs | 6 |
| Mini batch size | 80 |
| Initial learn rate | 0.0005 |
| Weight learn rate factor | 10 |
| Bias learn rate factor | 10 |
| Learn rate drop period | Piecewise |

DDR4 RAM. The procedures for image transformation, model implementation, and validation were developed in MATLAB 2019a (MathWorks, Inc., Natick, MA, USA); the coding scripts are available at Zenodo, DOI: 10.5281/zenodo.7384664. Table 2 details the transfer learning training parameters used in this work, which were obtained from *Altuntaş, Cömert & Kocamaz (2019)* and *Chen et al. (2021b)* and manually adjusted.

# RESULTS AND DISCUSSIONS

## Image dataset

We created a dataset containing seventeen thousand images, one thousand per class, in JPG format and derived from these an augmented dataset of sixty-eight thousand images containing converted images to the four previously commented color spaces. Other works have been developing efforts to implement databases according to the objectives of each particular investigation, such as food databases (*Ciocca, Napoletano & Schettini, 2018*) and seeds (*Espejo-Garcia et al., 2020*); however, these databases are far from generalized. This could be due to the semantic difficulties of organizing classes in a table that mean the same thing for different classifiers (*Ciocca, Napoletano & Schettini, 2018*). As an example, Fig. 5 shows the first element for each class in the five-color spaces used for training.

The pretreatment focused on conversions of full image datasets from RGB to four color spaces, preserving details and maintaining a balanced number of elements per group, which helps in the proper training of the neural network (*Ciocca, Napoletano & Schettini, 2018*; *Blok et al., 2022*). This is similar to what was performed by *Castro et al. (2019a)* who, using RGB images, extracted mean values and converted them to another $L^*a^*b^*$ and HSV in a previous stage of classifiers training, obtaining that the best combination was supported vector machine and RGB color space. Although it is clearly different from works such as those followed by *Abade et al. (2022)*, *Lin et al. (2018)*, *Chen et al. (2020)* and *Chen et al. (2021b)*, which increased the dataset by image manipulation, or *Moses et al. (2022)* who used high-magnification images datasets of damaged rice grains in order to evaluate classifiers performance; who explains that pixel information could be limited, and another way to increase it should be studied.

## Training transfer

In order to follow the training progress using the image dataset in different color spaces for each metric, their main statistical metrics *accuracy* and *loss* are plotted; see Fig. 6.
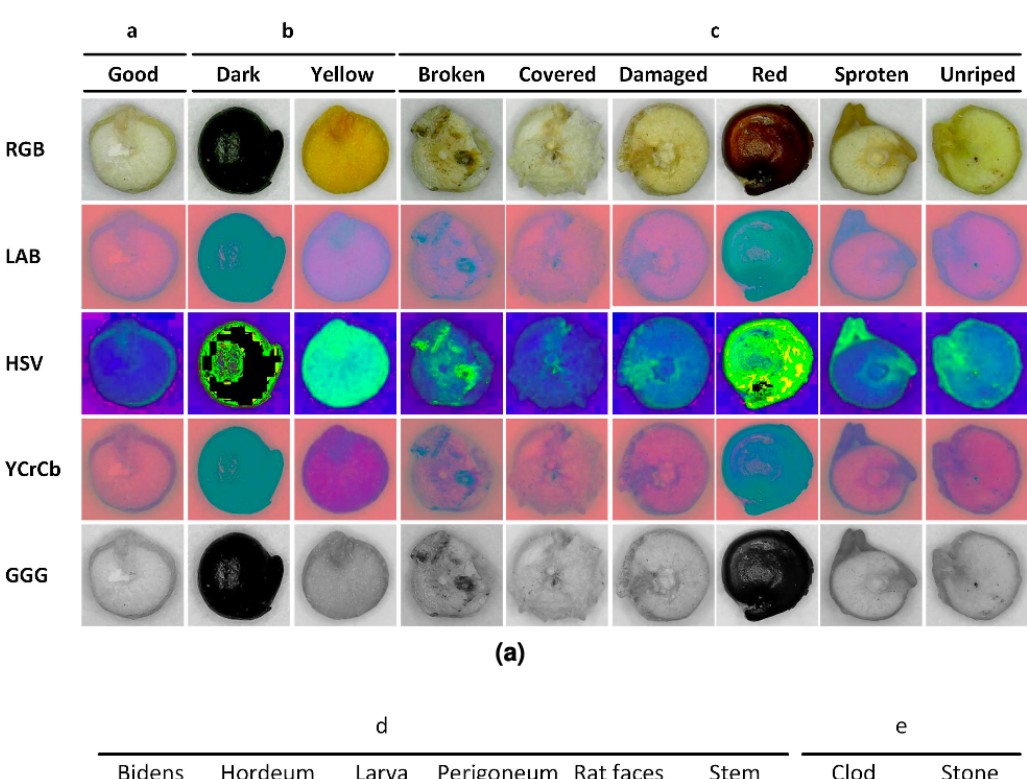

**(a)**

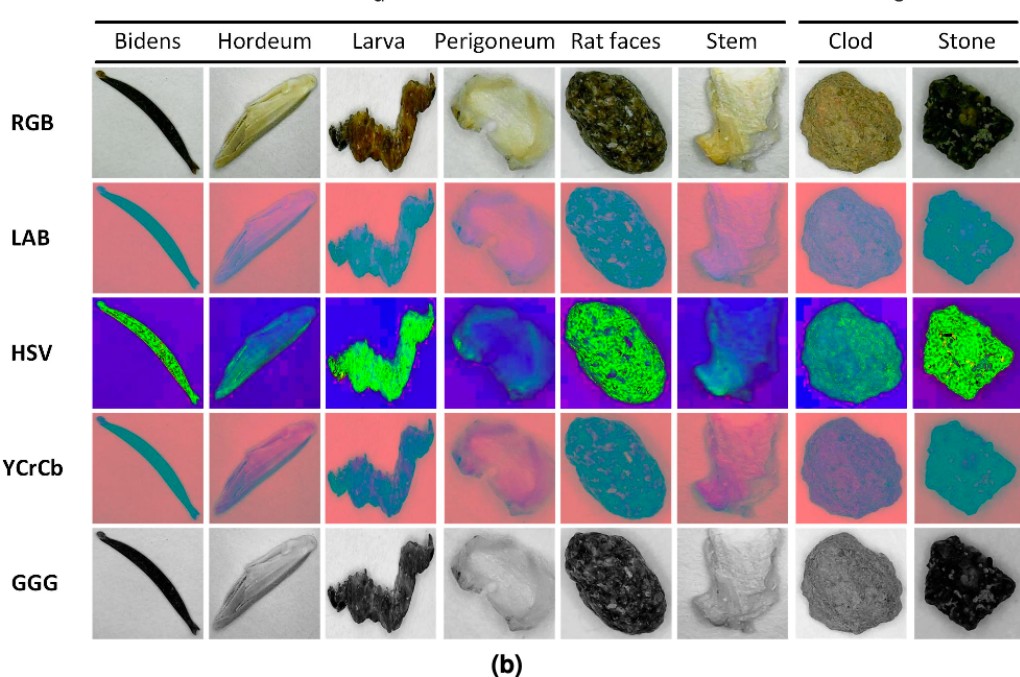

**(b)**

**Figure 5** (A–B) Examples of images per class and color space.

The accuracy reached for all models during training was over 95%, results comparatively similar to other works ad *Miraei Ashtiani et al. (2021)* when reduced size products or their contaminants are discriminated. In the same figure, it is observed, with the exception of *AlexNet*, that there was no effect of color space on the statistical metric of model training;

this could be due to improvements in the generalization capacity of new models such as *Movilenet-V2* and *DenseNet-201* as a comment (*Espejo-Garcia et al., 2020*). On the other hand, AlexNet model metrics showed better results when the RGB color space dataset was used. This is contrary to the results reported by *Castro et al. (2019a)*, who trained classifiers with different color spaces and obtained better the low results with dataset in L*a*b* color space; however, this could be because the models were trained from scratch and pre-trained models based on RGB images were used in the present study.

Likewise, according to the graphs of the average values, it is observed that the *DenseNet-201* network is significantly better than the MobileNet and AlexNet. The latter is slightly better in training with less than 300 epochs, subsequently equaling his training metrics. What is more, according to the convergence of the precision and loss curves, it follows that they are inversely related to the complexity of the model, consistent with the proposed by *Too et al. (2019)*, *Chen et al. (2021b)*, *Ciocca, Napoletano & Schettini (2018)* and *Miraei Ashtiani et al. (2021)*.

Secondly, the number of iterations required to find stability in our work, approximately five hundred, is significantly high compared to that reported by *Abade et al. (2022)* one hundred, *Altuntaş, Cömert & Kocamaz (2019)* sixty, or *Too et al. (2019)* thirty iterations. This is likely related to the number of categories involved in the experiments, five, two, or eight classes, respectively.

Figure 7 shows the mean confusion matrices for each combination of convolutional neural network and color space. It is observed, in all cases, that discrimination capacities were above 95%. Also, it is observed that the use of image databases in different color spaces did not strongly influence the accuracy of the convolutional neural networks. By another hand, classification errors were grouped mainly in contrast and faulty grain, organic and inorganic materials, according to previously done in Fig. 2.

Although recent research focuses on evaluating new architectures, prioritizing features such as generalization ability and lower complexity, among others, *Espejo-Garcia et al. (2020)*; some examples are the works of *Duong et al. (2020)* for fruit recognition, *Ciocca, Napoletano & Schettini (2018)* for food in general. Nevertheless, the main critique for this kind of work in machine learning application is clarity and lack of the criteria to establish structural elements *Castro et al. (2017)*, a problem that has been adequately explained and addressed by *Dsouza, Huang & Yeh (2020)* using combinatorial experiments.

In our case, to compare the computational cost for training transfer of three well-known structures, from less complex (AlexNet) to more complex (DenseNet-201) and one optimized for mobile applications (MobileNetV2), the consumed-time was determined and plotted in Fig. 8.

The comparing base was the training-transfer time for the combination of AlexNet-RGB color space; so it was noticed that in the mean MobileNetV2 and DenseNet-201 required over 1.43 and 8.57 times the used by AlexNet, concordant with their rising structural complexity.

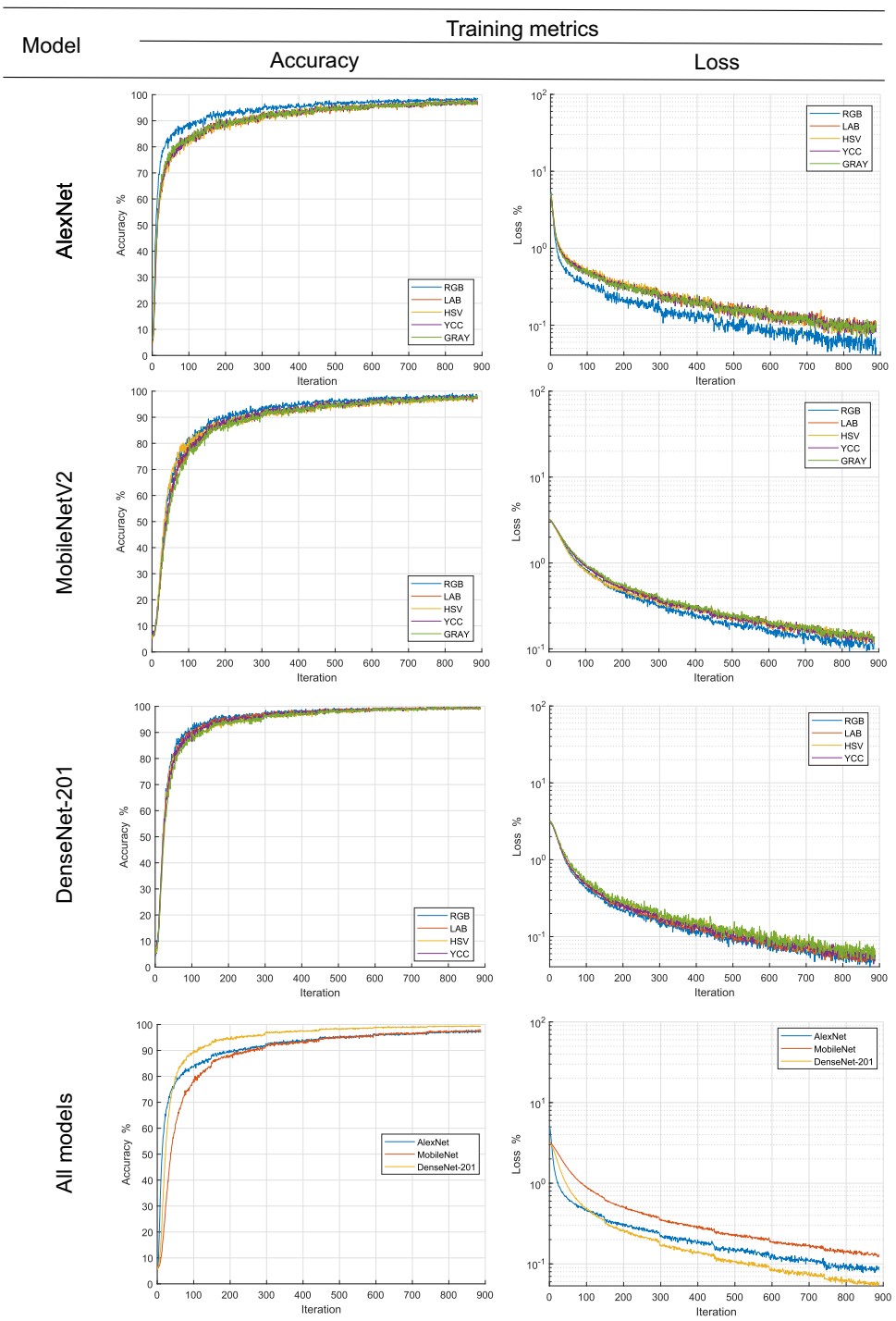

**Figure 6** Progress of the training stage.

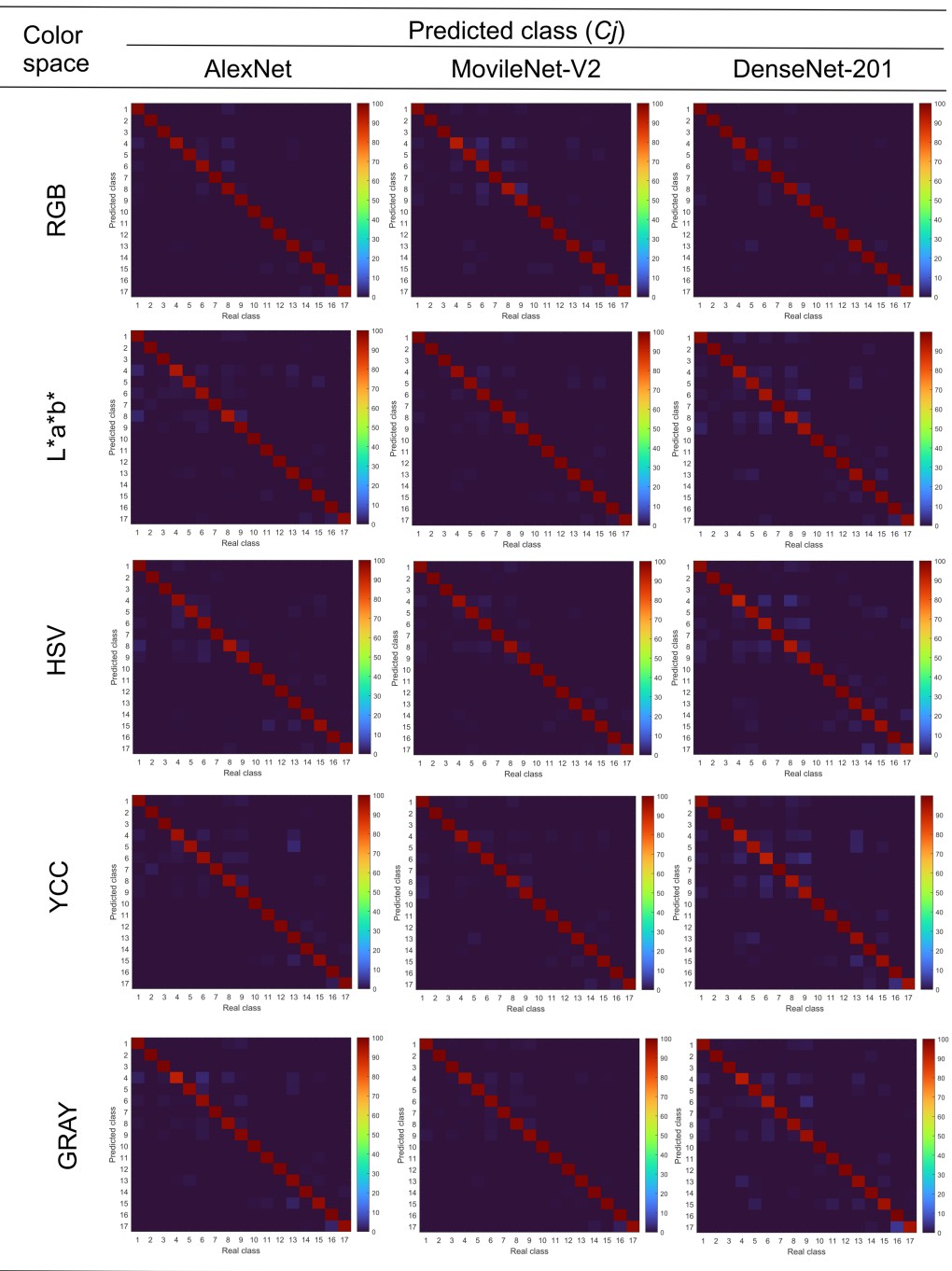

**Figure 7** **Mean confusion matrices of combinations CNN—color space.**

## Statistical metrics

Figure 9 and Table 3 show the *F-score* values for each *CNN - Color Space* combination. Other studies have focused on grain discrimination using some well-known networks and RGB images (*Li et al., 2021*; *Javanmardi et al., 2021*; *Huang et al., 2022*, among others) or

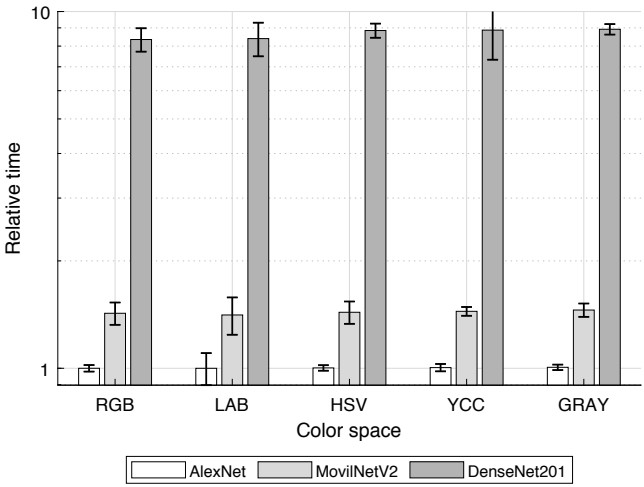

**Figure 8 Mean time for models training.**

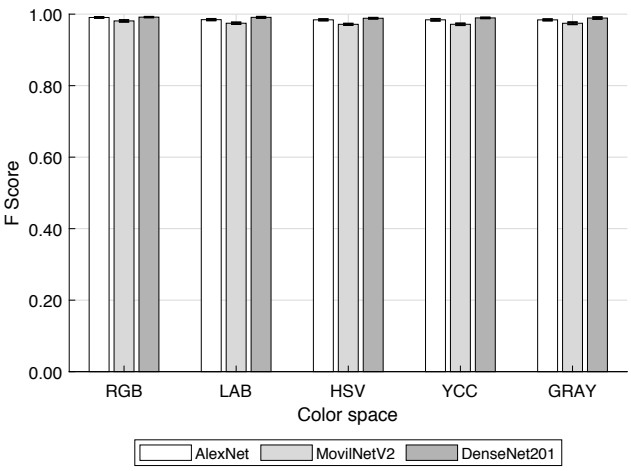

**Figure 9 F-score for each convolutional neural network and color space.**

more recently *THz* images (*Shen et al., 2021*). Nevertheless, the *F-score* obtained was similar in both types of images above 95%. Therefore, the use of complex technologies such as *THz* images would not significantly improve the classifiers. Consequently, at this level, it could be considered that the use of RGB images or their variants combined with convolutional neural networks is the most technically appropriate approach for discriminating visibly different elements.

Secondly, investigations such as that of *Li et al. (2021)* and *Huang et al. (2022)*, among others, have compared different convolutional neural network structures in grain discrimination showing that the complexity of the networks can improve their adaptability to different data in training transfer. In addition, despite not finding reports of the use of color spaces other than RGB to train networks by the previously mentioned method, so

**Table 3  F-scores per combination of CNN and color space.**

| Color space | CNN | | |
|---|---|---|---|
| | AlexNet | MobileNetV2 | DenseNet-201 |
| RGB | $0.991 \pm 0.002$ | $0.981 \pm 0.003$ | $0.992 \pm 0.001$ |
| LAB | $0.985 \pm 0.002$ | $0.975 \pm 0.003$ | $0.991 \pm 0.002$ |
| HSV | $0.984 \pm 0.002$ | $0.972 \pm 0.002$ | $0.989 \pm 0.002$ |
| YCC | $0.984 \pm 0.003$ | $0.972 \pm 0.003$ | $0.990 \pm 0.002$ |
| Gray | $0.984 \pm 0.002$ | $0.975 \pm 0.003$ | $0.989 \pm 0.001$ |

**Table 4  Summary of multiple range test for the CNNs.**

| Source | Squared sum | DoF | Mean square | F-score | P-value |
|---|---|---|---|---|---|
| Main effects | | | | | |
| A-CNN | 0.0124 | 2 | 0.0062 | 1163.79 | 0.0000 |
| B-Color space | 0.0160 | 4 | 0.0004 | 75.06 | 0.0000 |
| Interaction | | | | | |
| AB | 0.0004 | 8 | 0.0001 | 9.37 | 0.0000 |
| Residues | 0.0015 | 285 | 0.0000 | | |
| Total (corrected) | 0.0159 | 299 | | | |

**Notes.**
F-scores are based on mean square of residual error.

the adaptability of networks of greater complexity reduces the effect of changing the color space during training.

The results show how the architectures used in this study were capable of being trained using not only RGB images but, at the same time, a reduction in F-score was observed when other color spaces were used. In this sense, the results of the multi-factorial *ANOVA* test are summarized in Table 4, it is according to $P$-value $< 0.01$ explain that either CNN and color space have a statistical effect on F-score with a 99% of confidence level.

Likewise, by mean the Tukey's test of multiple the *F-score* was grouped according to color space influence on it, these groups are (a) RGB, (b) *HSV, YCC, and GRAY*, (c) *LAB, GRAY*. Initially, these architectures were created and trained using RGB images for specific cases and extended to other case studies (*Abade et al., 2022*; *Ciocca, Napoletano & Schettini, 2018*). Nevertheless, as it is being demonstrated, only in the case that the images to be used are images in the RGB color space would it be expected to obtain the maximum discrimination efficiency. Also, it can be seen that when networks optimized for specific purposes, such as the MobileNetV2 network, are used, the loss of efficiency due to training with other color spaces is more pronounced compared to the DenseNet-201 network. This is coherent with *Ciocca, Napoletano & Schettini (2018)*, who commented that when databases are compared for CNN training, the accuracy depends on the representativeness of the database.

## CONCLUSIONS

All tested CNNs showed an F-score from 98 to 99% for quinoa grains and foreign bodies discrimination. Statistically, both color space and CNN structure produce significant effects over F-score for discrimination of foreign bodies in quinoa grains. In this study, DenseNet-201 was the most robust, with a reduced effect of color space on its performance. But, at the same time, this CNN requires more than eight times the time for training transference than AlexNet. Although, for all the cases, these slight differences in accuracy among the studied CNNs would not affect its practical applications for foreign bodies' discrimination in quinoa grains. Regarding the color spaces, in general, the RGB provided the means to obtain the highest F-scores regardless of the method; this may follow the models' hyperparameter tuning, which usually considers the most common color space. Considering further research directions, and given the expected small proportion of foreign bodies compared to the true quinoa grains, it may be relevant to employ adaptive and skew-sensitive techniques to reduce the bias towards the positive class.

## ACKNOWLEDGEMENTS

The authors thank the National University de Frontera, which provided the equipment used through the project "Creation of the food safety research laboratory service at the Universidad Nacional de Frontera with CUI No 2439545".

### Funding

This work was supported by Universidad Nacional de Frontera (CUI No. 2439545). The funders had no role in study design, data collection and analysis, decision to publish, or preparation of the manuscript.

### Grant Disclosures

The following grant information was disclosed by the authors:
The Universidad Nacional de Frontera: CUI No. 2439545.

### Competing Interests

The authors declare there are no competing interests.

### Author Contributions

- Himer Avila-George conceived and designed the experiments, analyzed the data, prepared figures and/or tables, authored or reviewed drafts of the article, and approved the final draft.
- Miguel De-la-Torre conceived and designed the experiments, authored or reviewed drafts of the article, and approved the final draft.
- Jorge Sánchez-Garcés performed the experiments, prepared figures and/or tables, authored or reviewed drafts of the article, and approved the final draft.

- Joel Jerson Coaquira Quispe analyzed the data, authored or reviewed drafts of the article, and approved the final draft.
- Jose Manuel Prieto analyzed the data, authored or reviewed drafts of the article, and approved the final draft.
- Wilson Castro conceived and designed the experiments, performed the experiments, prepared figures and/or tables, authored or reviewed drafts of the article, and approved the final draft.

### Data Availability

The dode and data are available at Zenodo: Himer Avila-George. (2022). himerag/quinioa: Initial release (1.0). Zenodo. https://doi.org/10.5281/zenodo.7384664.

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
