# Peer review of "Discrimination of foreign bodies in quinoa (Chenopodium quinoa Willd.) grains using convolutional neural networks with a transfer learning approach"

_PeerJ, doi:10.7717/peerj.14808_

## Round 0.1 · original submission · Minor Revisions

The main concerns that the reviewers have expressed are in the presentation of results and selection of the test dataset. The train-test-valid split is not apparent in the data and as a result, reviewer #2 has raised a concern regarding the selection of test data and hyperparameter selection. Please address these concerns in your re-submission. Reviewer #3's concerns on explaining in detail the color spaces used would definitely add value to the paper. Please consider adding figure(s) to highlight how the grains/noise classification is affected in specific examples in different color spaces (examples taken from misclassified samples from the confusion matrix). Also, please consider adding text in discussion based on Reviewer #1's suggestions on prior work in this area. There are several other minor points raised by each of the reviewers on presentation of the text and citations, please correct these minor issues before the re-submission.

·

Basic reporting

no comment

Experimental design

no comment

Validity of the findings

no comment

Additional comments

This paper presents the classification of different impurities from quinoa grains using three deep-learning algorithms, including Alextnet, MobileNetv2, and DenseNet-201. This study demonstrates that the CNN classifier is an efficient tool for the intelligent discrimination of foreign bodies in quinoa grains. The article has a proper structure and provides valuable information to the readers. However, it needs some modifications. My few minor concerns are:
Page 5, line 140: A more precise definition of transfer learning should be provided. The following article is suggested in this regard, which has well presented the hidden mechanisms in this process: https://doi.org/10.1109/ACCESS.2021.3096550
Page 7, line 168: Splitting the dataset into training and validation without a test subset should be justified by the authors (without substantive change, i.e., without changing the structure of the paper). The mention cause of this issue is rarely stated in articles. The authors can find the technical reason in the above-suggested publication.
Page 10: The superiority of one model in comparison to another should be justified taking into account the structure and topologies of the CNN models.
Results and discussion section: Please compare the results obtained in the present research with other research conducted on seeds and grains to develop the discussion section. You can refer to the following recent publications:
https://peerj.com/articles/cs-639/
https://doi.org/10.1016/j.jspr.2021.101800
https://doi.org/10.1016/j.compag.2020.105931
https://doi.org/10.1016/j.compag.2022.107393

·

Basic reporting

Summary:
This paper presents an empirical study of pretrained convolutional networks for identifying foreign bodies in quinoa samples. Quinoa samples were collected and manually divided into 17 classes: quinoa grains and sixteen foreign bodies. Then, a thousand images for each type of quinoa are annotated manually and considered for training a classifier using three pretrained convolution networks: AlexNet, MobileNet-V2, and DensNet-201 with different color spaces. Results conclude that DenseNet-201 with RGB color space performs best among others, yet it is the most time-consuming in training and inferencing.

Following are my comments:
1. Authors should write more about the application of image-based foreign body identification in quinoa samples.

2. Minor writing issues
a) What is b_i in equation 1 ? that should be in paper
b) Equation 3 is not written correctly. Please see the correct softmax formula from https://en.wikipedia.org/wiki/Softmax_function
c) What is n in equations 5,6,7,8. should be written on paper

Experimental design

1. Many palace's correct references are missing:
AlexNet: ImageNet Classification with Deep Convolutional Neural Networks, NIPS-2012
MobileNetV2: Inverted Residuals and Linear Bottlenecks, CVPR 2018
DensNet: Densely Connected Convolutional Networks, CVPR 2017


2. Details missing:
a) What is the size of the test set and how it is obtained?
b) How the values of the hyperparameters are obtained in table 2?
c) How to obtain the best checkpoint?

Validity of the findings

The proposed approach is sound and experiments are concrete. Authors should release the dataset for reproducibility.

Reviewer 3 ·

Basic reporting

The work presented is clear and easy to read. The references are current and are in line with the objective of the work. However, adjustments to the tables must be made, as tables 3 and 4 are far from their explanation texts. The structure of the work is adequate and well structured, aligning the Introduction with the basic literature used to discuss the results obtained with the experiments. Figures and tables are necessary and help to discuss the results.
Thus, basic adjustments must be made before acceptance.

Experimental design

Research is original and in line with the objectives and scope of the journal. The line of research is well defined, relevant and will bring good contributions to the area under study.
The authors sought to discuss the main points according to the objectives outlined, using current literature to discuss the results obtained with the experiment, as well as clearly describing the methodology used, as well as a number of images well above the expected for the application in study.

Validity of the findings

The data presented by the performance of the CNN models used are clear and well defined in tables 3 and 4. However, in table 4 the authors could have identified the type of test used to predict that the models do not differ from each other, although ANOVA does not have observed differences between them.
The authors clearly discussed the results based on current literature, which highlights the importance of the work for the area under study, as well as the relevance of the studies carried out involving machine learning.
The work is clear and well-designed. However, positioning adjustments from tables 3 and 4 must be corrected. Another important aspect that would bring a greater contribution to the area would be a greater discussion between the color models used and the performance obtained by the learning models.
The conclusion is clear and responds to the objectives outlined.

Additional comments

The work is clear and well-designed. However, positioning adjustments from tables 3 and 4 must be corrected. Another important aspect that would bring a greater contribution to the area would be a greater discussion between the color models used and the performance obtained by the learning models.

---

## Round 0.2 · Minor Revisions

Thank you for addressing all the reviewer's comments. As such the manuscript is ready for acceptance. However, I see that Reviewer #2's comments on data availability has been addressed in the rebuttal letter. But, the dataset link is not in the manuscript. Please add the link to the dataset either at end of Introduction or in the Methods section where the data set creation is described. Apart from this change, there are no other changes required for accepting the paper.

---

## Round 0.3 · accepted · Accept

The manuscript is now ready for publication.